# Resonance Searches with Machine Learned Likelihood Ratios

**Jacob Hollingsworth*** **and Daniel Whiteson**

Department of Physics and Astronomy, University of California, Irvine, CA 92697, USA
* jhollin1@uci.edu

October 19, 2020

## Abstract

We demonstrate the power of machine-learned likelihood ratios for resonance searches in a benchmark model featuring a heavy $Z'$ boson. The likelihood ratio is expressed as a function of multivariate detector level observables, but rather than being calculated explicitly as in matrix-element-based approaches, it is learned from a joint likelihood ratio which depends on latent information from simulated samples. We show that bounds drawn using the machine learned likelihood ratio are tighter than those drawn using a likelihood ratio calculated from histograms.

## 1 Introduction

A primary focus of the physics program at the Large Hadron Collider is the search for beyond the standard model (BSM) physics. Many BSM models predict the existence of heavy, short lived particles that rapidly decay to familiar standard model particles, leaving a telling experimental signature known as a resonance. Historically, the discovery of new resonances has revolutionized our understanding and confidence in models of particle

physics [1, 2, 3, 4, 5, 6, 7]. A similar discovery made in the current age would likely have a similar impact.

Many searches for BSM resonances involve fitting signal and background spectra in the invariant mass of the final state particles, allowing one to construct a likelihood ratio [8, 9, 10, 11]. Recently, this has been greatly improved by utilizing machine learning at various steps in the process [12, 13]. However, relying solely on the invariant mass neglects a great deal of the available information when determining the likelihood ratio of an event. The full event information is given by the set of four-momenta of every particle in the final state. Summarizing this relatively high dimensional information with invariant mass plausibly results in a significant amount of information loss.

Certain methods have been developed that aim to overcome this information loss [14, 15, 16, 17, 18, 19]. The matrix-element method searches for new particles by approximating the detector response with a simple transfer function and marginalizing matrix elements over the unseen detector degrees of freedom [20, 21, 22]. This approach uses multivariate detector level output in determing the likelihood ratio, but critically relies on the choice of transfer functions and can often be very computationally expensive to perform.

In this paper, we implement a new method of analysis, originally applied to effective field theories (EFTs), which machine learns a likelihood ratio from latent information that is extracted from simulations [23, 24, 25, 26]. Similar to the matrix element method, this new method utilizes multivariate output of a detected event and so it is expected to provide similar improvements in performance. However, it removes the need to approximate the detector using transfer functions, and thus may be viewed as a direct improvement over the matrix element method.

Transferring the methods of Refs. [23, 24, 25, 26] from EFTs to resonance searches is non-trivial for several reasons. First, EFTs have a morphing structure, allowing squared matrix elements to be written as simple polynomials of the parameters of the theory [23, 27]. Consequently, by evaluating the squared matrix element of an event at a handful of "benchmark" points in the parameter space, one is able to infer the squared matrix element for arbitrary theory parameters, affording significant improvements in computational efficiency. However, resonanace searches do not have a complete morphing structure, as the squared matrix element typically has a non-polynomial dependence on mass. As such, the squared matrix element of an event must be evaluated at every point in the parameter space.

Additionally, the most successful methods for EFTs rely on using the derivative of the log likelihood ratio with respect to the theory parameters, known as the score, to successfully train the machine learning models. Scores are readily available in EFT calculations, but must be numerically calculated for resonance searches, which greatly increases the computational cost of the analysis. As a result, we are constrained to use the less sample efficient methods, which do not rely on this information during training. This work is the first to show that these difficulties can be overcome, and that these methods may still provide substantial improvements over traditional methods beyond the context of EFTs.

In Section II, we review the theory behind performing a resonance search and review the new method that we use to calculate the likelihood ratio. In Section III, we will introduce a simple BSM model and detail how the new framework will be used to search for a resonance in the model. In Section IV, we show the results of our resonance search. In Section V, we discuss the implications of this work and possible future directions.

## 2 Method

Let $\theta$ denote a set of theory parameters. Using a standard suite of programs, we can produce a set of simulated events $\mathcal{X} = \{x \sim p(x|\theta)\}$, where $x$ is a random variable of detector level observables and is used interchangably as a realization of this random variable. However, the inverse problem of determining which $\theta$ produced a set of events $\mathcal{X}$ is extremely difficult. The Neyman-Pearson lemma shows that the optimal discriminator between two competing theories, parameterized by $\theta$ and $\theta_0$ respectively, is the likelihood ratio:

$$r(x|\theta, \theta_0) = \frac{p(x|\theta)}{p(x|\theta_0)} = \frac{\int dz\, p(x|z)p(z|\theta)}{\int dz\, p(x|z)p(z|\theta_0)}, \tag{2.1}$$

where $z$ is the latent parton level four momenta of all particles in the final state, which is unobservable in experiment.

For typical showering and detector simulations, $p(x|z)$ is intractible due to the extremely large number of ways an event may shower and be detected. For this reason, $r(x|\theta, \theta_0)$ is typically calculated by attempting to approximate $p(x|\theta)$ and $p(x|\theta_0)$ directly, using histograms of $x$. The number of data points required to adequately populate the bins of the histogram scales exponentially with the dimension of $x$, which is known as the curse of dimensionality. As a result, this approach becomes impractical as the dimension of $x$ becomes moderately large. As such, histograms usually are constructed in only one or two summary statistics of $x$, which may result in information loss relative to higher dimensional approaches.

A recent study with effective field theories has offered an alternative method of calculating $r(x|\theta, \theta_0)$ as a function of the detector level output [23]. The remainder of this section follows the discussions in Refs. [23, 24, 25, 26] very closely. We first consider the joint likelihood ratio:

$$r(x, z|\theta, \theta_0) = \frac{p(x|z)p(z|\theta)}{p(x|z)p(z|\theta_0)} = \frac{p(z|\theta)}{p(z|\theta_0)}. \tag{2.2}$$

All intractable parts of the joint likelihood ratio cancel, and for simulated events, we can calculate $r(x, z|\theta, \theta_0)$ in terms of tractible matrix elements as

$$r(x, z|\theta, \theta_0) = \frac{\sigma(\theta)^{-1}|\mathcal{M}(z|\theta)|^2}{\sigma(\theta_0)^{-1}|\mathcal{M}(z|\theta_0)|^2}, \tag{2.3}$$

where $\sigma$ gives the cross section as a function of theory parameters and $\mathcal{M}$ gives the matrix element as a function of parton level momenta and theory parameters. The joint likelihood ratio cannot be calculated for observed data, as it requires knowledge of $z$ which is not well defined in experiment. However, in the following paragraphs, we demonstrate that machine learning the joint likelihood ratio of simulated events will result in the true likelihood ratio in Eq. (2.1) under a specific set of conditions.

Consider a function $\hat{r}(x|\theta, \theta_0)$ that attempts to predict $r(x, z|\theta, \theta_0)$ given only information of $x$. We can quantify the error of the function when evaluated on a set of data $(x, z) \sim p_{train}(x, z)$ with the functional:

$$\mathcal{L}[\hat{r}] = \int \int dx\, dz\, p_{train}(x, z)\, |\hat{r}(x|\theta, \theta_0) - r(x, z|\theta, \theta_0)|^2. \tag{2.4}$$

It can be shown that minimizing this loss functional with data drawn from $p_{train}(x, z) = p(x, z|\theta_0)$ will yield the desired likelihood ratio. Thus, using a deep neural network to

represent $\hat{r}(x|\theta, \theta_0)$, we can use standard optimization techniques employed in machine learning to train an estimator that will converge such that

$$\hat{r}(x|\theta, \theta_0) = \frac{p(x|\theta)}{p(x|\theta_0)} \tag{2.5}$$

in the limit of infinite data, an infinitely large neural network, and perfect loss optimization. In realistic implementations, deviations from the true likelihood ratio may occur due to the effect of finite datasets, finite neural networks, and inefficient optimization. This likelihood depends only on $x$, and so it can be evaluated for simulated and observed events alike.

We can use the joint likelihood ratio to construct alternative, more complicated loss functionals that similarly converge to the true likelihood ratio. A summary of these methods as well as their different properties is found in the references [23, 25]. In this work, we find the $\hat{s}(x|\theta, \theta_0)$ that minimizes the ALICE (approximate likelihood with improved cross-entropy estimator) loss functional, which is given by:

$$\mathcal{L}[\hat{s}] = -\int\int dx\,dz\, p(x, z|\theta_0)\big[s(x, z|\theta, \theta_0)\log(\hat{s}(x|\theta, \theta_0)) +$$
$$(1 - s(x, z|\theta, \theta_0))\log(1 - \hat{s}(x|\theta, \theta_0))\big], \tag{2.6}$$

where we have defined

$$s(x, z|\theta, \theta_0) = \frac{1}{1 + r(x, z|\theta, \theta_0)}. \tag{2.7}$$

With $\hat{s}(x|\theta, \theta_0)$, we can form an estimator for the likelihood ratio

$$\hat{r}(x|\theta, \theta_0) = \frac{1 - \hat{s}(x|\theta, \theta_0)}{\hat{s}(x|\theta, \theta_0)}, \tag{2.8}$$

which similarly converges to the true likelihood ratio. The minimization is performed over a balanced training set, with an equal number of events drawn from $p(x|\theta)$ and $p(x|\theta_0)$.

Given the likelihood ratio of each event, the likelihood ratio over $\mathcal{X}$ is trivially found:

$$r(\mathcal{X}, \theta, \theta_0) = \prod_{x\in\mathcal{X}} r(x|\theta, \theta_0) \tag{2.9}$$

There are a handful of ways to diagnose mismodelling and poor convergence. Errors that result from simulations mismodelling the physical process can be addressed by introducing nuisance parameters and profiling over them [26, 28]. Errors due to poor convergence of the neural network can be addressed by increasing network complexity, increasing the training set size, or tuning learning parameters. Among other methods, poor convergence can be diagnosed by extensively sampling many $p(x|\theta)$ and $p(x|\theta_0)$ and visually comparing the distributions $p(x|\theta)$ and $\hat{r}(x|\theta, \theta_0)p(x|\theta_0)$. However, in higher dimensions, it may be necessary to compare distributions of summary statistics instead [23, 28].

## 3 Experiment

We consider collisions $q\bar{q} \to q\bar{q}$ in the standard model with a massive $Z'$ boson included [29, 30]. The $\theta$ that parameterizes this theory is $\theta = (M_{Z'}, g_{Z'})$, where $M_{Z'}$ is the mass

Table 1: Table of hyperparamaters used to the train neural network

| Hyperparameter | Value |
| --- | --- |
| Activation function | tanh |
| Number of hidden layers | 3 |
| Neurons per hidden layer | 12 |
| Initial learning rate | $2.2 \times 10^{-3}$ |
| Final learning rate | $10^{-4}$ |
| Learning rate decay schedule | Linear |
| Optimizer | AMSGrad |
| Batch Size | 128 |
| Validation Split | .25 |
| Number of Epochs | 100 |
| Training Samples (unweighted) | $10^6$ |
| $\theta_0$ | (300 GeV, 2.0) |

of the $Z'$ boson and $g_{Z'}$ is its coupling to standard model quarks. We attempt to find the $Z'$ resonance using two different methods of calculating the likelihood ratio: the ALICE approach described above and the benchmark histogram-based approach, utilized in many current LHC searches [31, 32, 33, 34].

We sample $10^4$ events at every point on a grid in $\theta$-space spanning $M_{Z'} \in [275, 325]$ GeV and $g_{Z'} \in [0, 2]$, with grid spacing $\Delta M_{Z'} = 5$ GeV, $\Delta g_{Z'} = .2$, yielding a total of $1.21 \times 10^6$ weighted events. We use a constant width $\Gamma_{Z'} = 2.4$ GeV for every theory in the grid. Events are sampled using MadMiner [26], which generates, showers, and detects events using MadGraph v2.6.5, Pythia8 and Delphes, respectively [35, 36, 37]. Jets are reconstructed using the anti-$k_T$ algorithm with distance parameter $R = .5$ [38]. Events where more or fewer than two jets are detected are discarded.

For each event, our observables $x$ consist of the four-vector of each reconstructed jet. A cut is placed on the invariant mass so that $m_{jj} \in [150, 450]$ GeV and on the jet transverse momenta such that $p_T > 20$ GeV. For each sample, we calculate the tree-level joint likelihood ratio at every other grid point, which MadMiner performs by using Madgraph's reweight feature [39]. The joint likelihood ratios are then used to unweight the samples in preparation for machine learning.

We focus on a qualitative assessment of the application of machine learned likelihood ratios to resonance searches. As such, we neglect additional complications that may arise in an experimental setting that are not expected to affect qualitative results, such as pile up interactions, trigger strategies, the dependence of $\Gamma_{Z'}$ on $M_{Z'}$, or additional diagrams that may contribute to the detected final state. An interesting direction for future study would be to incorporate these effects into the analysis.

We use MadMiner to train a neural network capable of calculating $r(x|\theta, \theta_0)$ using the ALICE loss functional. We parameterize the dependence on $\theta$, as described in Refs. [23, 40], which allows us to evaluate $r(x|\theta, \theta_0)$ at any $\theta$ in the parameter space using a single neural network. The neural network is trained with hyperparameters given in Table 1 to minimize the loss in Equation 2.6. We fix $\theta_0 = (300 \text{ GeV}, 2.0)$, though results should be independent of this choice.

We also calculate the likelihood ratio from histograms. For this method, we bin events sampled from $p(x|\theta)$ and $p(x|\theta_0)$ in invariant mass, and the likelihood ratio is again given by the ratio of normalized counts in each bin. We use a fixed bin size of 20 GeV at all points in the parameter space.

We separately generate a test dataset $\mathcal{X}_{\text{test}}$ of 10000 events with $\theta_{\text{test}} = (285 \text{ GeV}, 1.2)$

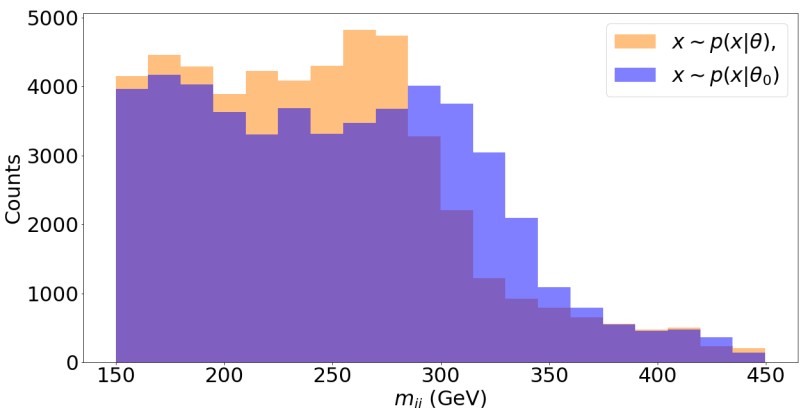

Figure 1: Distributions of invariant mass plotted for events drawn from $\theta = (285\,\text{GeV}, 1.0)$ (orange) and $\theta_0 = (315\,\text{GeV}, 1.0)$ (blue). We use these histograms to calculate the likelihood ratio, given by the ratio of counts in each bin.

and compare the results of a resonance search using our machine learned $\hat{r}(x|\theta, \theta_0)$ to one using $r(x|\theta, \theta_0)$ calculated from histograms. The test set is used to calculate expected p-values for $N$ test events in the asymptotic limit. In this work, we take $N = 50$. To demonstrate the limit setting abilities of these methods, we follow the example in Ref. [23]. We assume an asymptotically large test set and calculate

$$p_\theta = \exp\left(N\langle \log r(x|\theta, \theta_{\text{MLE}})\rangle_{x \in \mathcal{X}_{test}}\right), \tag{3.1}$$

where $\theta_{\text{MLE}}$ is the parameters of the maximum likelihood estimate, which is the $\theta$ that maximizes $r(\mathcal{X}_{\text{test}}, \theta, \theta_0)$. This expression takes a simple form because the dimension of our parameter space is two.

Before presenting the results to the full problem described above, we attempt to develop an intuition regarding all of the likelihood ratios discussed thus far. For this, we limit our analysis to the set of events drawn from $\theta = (285\,\text{GeV}, 1.0)$ and $\theta_0 = (315\,\text{GeV}, 1.0)$. In Figure 1, we show histograms in invariant mass for events drawn from $\theta$ and events drawn from $\theta_0$. The ratio of counts in each bin gives the likelihood ratio if the invariant mass is the only information available. The logarithm of this likelihood ratio is plotted as the grey line in Figure 2.

The next step is to compare the likelihood ratio calculated from histograms to the machine-learned likelihood ratio as well as the joint likelihood ratio, which benefits from parton-level information. A neural network is used to discriminate between events sampled from $\theta$ and $\theta_0$ by minimizing the ALICE loss functional. To compare this to the benchmark result, we show the expected value of the machine learned log likelihood ratio within each invariant mass bin. A full justification of this comparison is provided in the appendix. This expectation is plotted for events sampled from $p(x|\theta)$ (solid) and $p(x|\theta_0)$ (dashed) in blue. The neural network uses multivariate detector level information, which allows it to make more powerful predictions than the histogram based approach, which only uses invariant mass.

We also plot the expected log joint likelihood ratio within each invariant mass bin in magenta for events drawn from $p(x|\theta)$ (solid) and $p(x|\theta_0)$ (dashed). The expected joint likelihood ratio represents the most powerful expected likelihood ratios possible, as it requires knowledge of all considered parton level event information. We see that the machine learning approach is able to use the extra available information to modestly

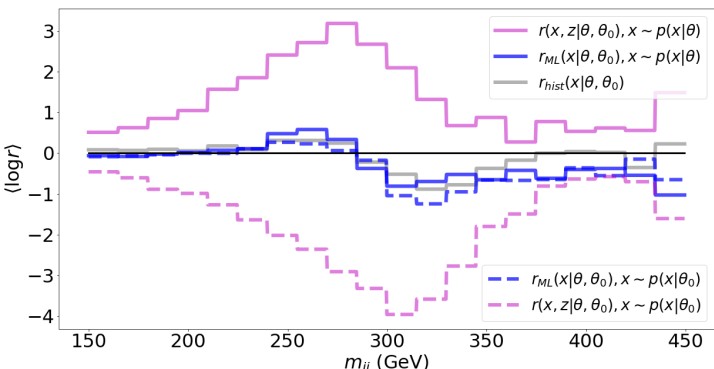

Figure 2: A comparison between the log likelihood ratio (grey, calculated using histograms in Figure 1), the expected machine learned log likelihood ratio (blue), and the expected log joint likelihood ratio (magenta) for events sampled from $\theta = (285 \text{ GeV}, 1.0)$ (solid) and $\theta_0 = (315 \text{ GeV}, 1.0)$ (dashed). Expectations are calculated with respect to all events that lie within the given invariant mass bin. We remark that the expected value of $r_{\text{hist}}(x|\theta, \theta_0)$ in a bin does not depend on the distribution from which an event is sampled, as only the number of events within each bin will change. The expected machine learned likelihood ratio is closer than the histogram approach to the expected joint likelihood ratio, which represents the optimal expected log likelihood ratio if given complete parton information of every event. The neural network approaches are not well converged at large invariant masses due to the lack of events in this region of the feature space.

outperform the histogram approach.

## 4    Results

In Figure 3, we plot p values as a function of mass and coupling, calculated using the ALICE likelihood ratio. In Figure 4, we show the same plot, calculated using histogram based likelihood ratios. While both approaches are capable of selecting the correct region of $\theta$-space, the results are significantly less constrained for the histogram approach than when using the ALICE based likelihood ratio.

We remark that only kinematic information is used to form the likelihood ratios used in Figures 3 and 4. A full analysis would include total rate information. However, the contribution of rate information to the log likelihood ratio is independent of the method used to calculate the kinematic portion of the log likelihood ratio, and thus is unable to change the relative ordering of the methods.

We believe that the ALICE likelihood ratio is able to outperform histogram based approaches due to the increased information utilized by ALICE. That is, the ALICE likelihood ratio uses information of the full four-momenta of both final state jets. On the other hand, the histogram based approach only has access to the invariant mass of the jet pair and cannot be extended to use additional observables due to the curse of dimensionality.

A possible critique of this interpretation is that the histogram based approach does not perform as well as ALICE because it requires use of a binned PDF, which may result in information loss. To address this concern, we also train a neural network using the ALICE loss functional to predict the likelihood ratio as a function of only invariant mass, instead

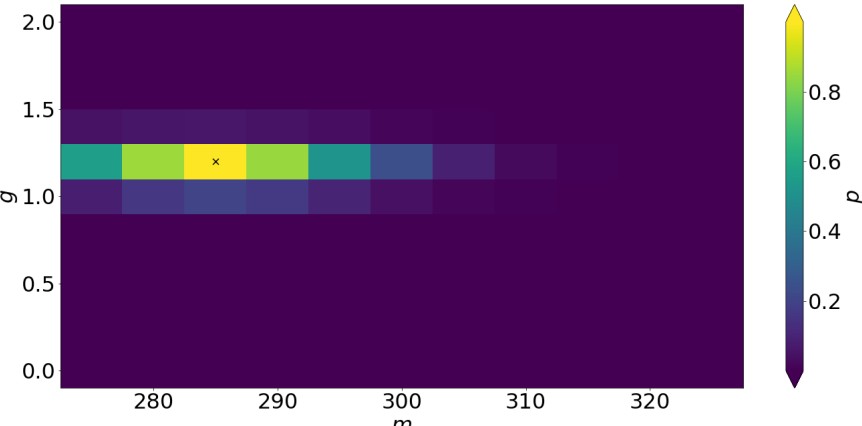

Figure 3: We show the expected p values plotted against mass and coupling for an Asimov test set drawn from the theory $(285 \text{ GeV}, 1.2)$. The p values are calculated using our machine learned likelihood ratio. The true value of $\theta$ is marked with a black x. In comparison to Figure 4, p values are more peaked around the true value of $\theta$.

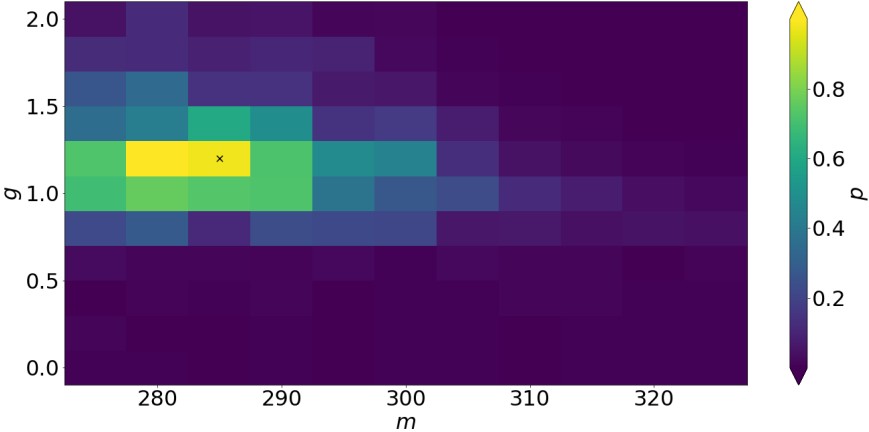

Figure 4: We show the expected p values plotted against mass and coupling for an Asimov test set drawn from the theory $(285 \text{ GeV}, 1.2)$. The p values are calculated using our likelihood ratio derived from histograms. The true value of $\theta$ is marked with a black x. In contrast to Figure 3, p values are more dispersed around the true value of $\theta$.

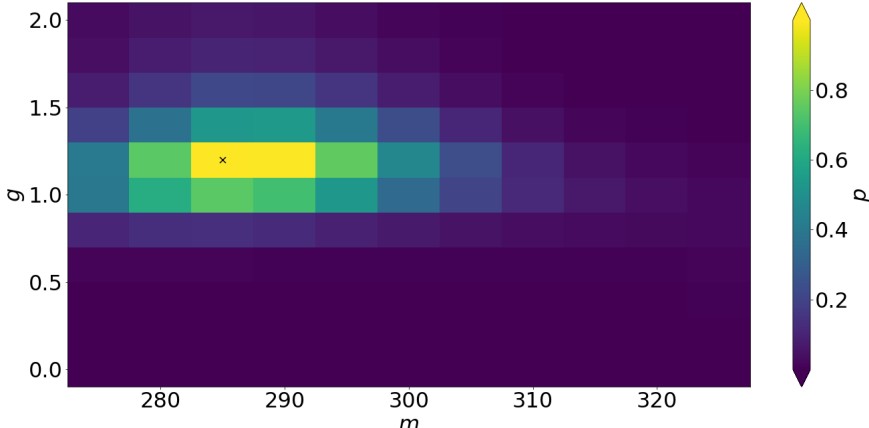

Figure 5: We show the expected p values plotted against mass and coupling for an Asimov test set drawn from the theory $(285 \text{ GeV}, 1.2)$. The p values are calculated using our machine learned likelihood ratio trained only on invariant mass. The true value of $\theta$ is marked with a black x.

of the full eight-dimensional dimensional jet four-momenta. This can be considered the continuous limit of the histogram based likelihood ratio, as it avoids the need for binning while only using the information in invariant mass. The expected p values plotted over $\theta$ are shown in Figure 5 and the hyperparameters used to train our neural network are shown in Table 2.

We see that the machine learned likelihood ratio trained only on invariant mass performs very similarly to the histogram result. This indicates there is not a substantial difference in available information due to binning effects.

In a final attempt to uncover the extra information used by the fully multivariate ALICE likelihood ratio, we train an additional neural network with the ALICE loss functional, but provide the detected invariant mass as well as the difference in pseudorapidity of the two final state jets, denoted $\Delta y_{jj}$. The input is thus two dimensional. The hyperparameters used to train the neural network are the same as those in Table 2, with the initial and final learning rates adjusted to $2.3 \times 10^{-3}$ and $4 \times 10^{-5}$, respectively.

Because the neural networks for the machine learned likelihoods include $\theta$ as an input, they offer a very natural interpolation in $\theta$-space relative to the histogram approach. In Figure 6, we use this property to compare the exclusion contours of the fully multivariate, eight dimensional machine learned likelihood ratio (blue), the two dimensional machine learned likelihood ratio (green), and the machine learned likelihood ratio that is only dependent on invariant mass (purple). We see that the neural network trained on two dimensional input performs very similarly to the neural network trained on eight dimensional input, and that both of these significantly outperform the one dimensional approach.

This result demonstrates that the extra information used by the fully multivariate approach is largely or nearly entirely captured in $\Delta y_{jj}$. While these exclusion contours depend on the hyperparameter $N$ and will also change if we included rate information into the analysis, the relative ordering will not depend on these factors.

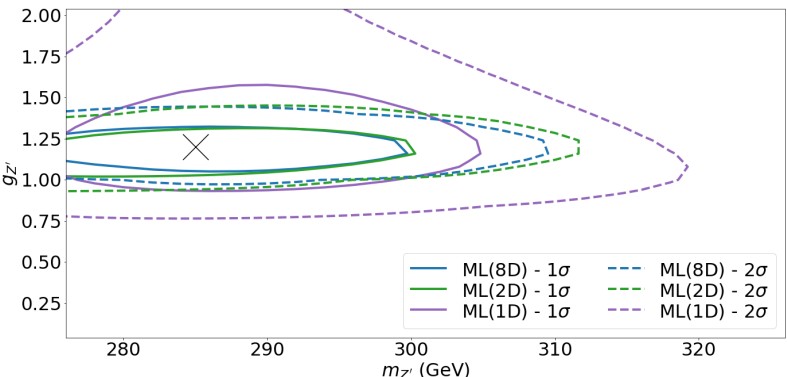

Figure 6: The $1\sigma$ (solid) and $2\sigma$ (dashed) exclusion contours for the machine learned likelihood ratio calculated using the full input (blue), invariant mass and $\Delta y_{jj}$ (green), and only invariant mass (purple). The black $x$ denotes the true value of $\theta$. The fully multivariate and two dimensional approaches perform very similarly, and both provide more powerful exclusion contours than the approach that uses only invariant mass.

Table 2: Table of hyperparamaters used to the train neural network with invariant mass as input.

| Hyperparameter | Value |
|---:|:---|
| Activation function | tanh |
| Number of hidden layers | 3 |
| Neurons per hidden layer | 8 |
| Initial learning rate | $10^{-3}$ |
| Final learning rate | $10^{-5}$ |
| Learning rate decay schedule | Linear |
| Optimizer | AMSGrad |
| Batch Size | 128 |
| Validation Split | .25 |
| Number of Epochs | 100 |
| Training Samples (unweighted) | $10^{6}$ |
| $\theta_0$ | $(300\ \mathrm{GeV}, 2.0)$ |

# 5 Conclusion

In this work, we have performed the first application of a novel likelihood-free inference method, ALICE, beyond the scope of effective field theories. ALICE, along with most methods from this new class of analysis techniques, relies on machine learning latent information extracted from simulations in order to produce a useful likelihood ratio. We have compared the new method to the traditional histogram based approach of performing a resonance search, and have seen dramatic improvement when multivariate detector level event information is included.

ALICE outperforms the histogram approach by providing significantly tighter exclusion contours. We believe that this improvement is similar to the improvement seen when using the matrix-element method, and originates from the greater amount of information that can be meaningfully utilized in multivariate analyses. Since we are now able to compare lower dimensional analyses to a fully dimensional analysis, we were able to conclude that nearly all information is captured in the observables $m_{jj}$ and $\Delta y_{jj}$ for our simple process. ALICE can be seen as an improvement over the matrix-element method, as it does not require one to approximate the detector using transfer functions.

Possibilities for future work include utilizing the partial morphing structure with the coupling to improve the computational efficiency of this work. One could also numerically evaluate derivatives of the joint likelihood ratio with respect to the mass. In combination with the partial morphing structure in the coupling, this would grant full access to the derivative of the joint likelihood ratio with respect to the theory parameters, which is known as the joint score. For EFTs, methods that include the joint score in the loss function have been more successful than those that neglect this information. It is possible that these results also generalize to resonance searches, and that even more information can be extracted from the detector level four-momenta.

Additionally, one could study if these results generalize to the case where a full treatment of systematic uncertainties in the input is performed. Since these methods scale well to high dimensional problems [41], one could also expand the space of observables to include the four-momenta of all jet constituents, rather than only using reconstructed jet four-momenta. This would potentially increase the information available to the neural network, at the cost of increasing the dimensionality of the problem. Finally, this work may be applied to more complicated resonant processes. We expect that, since we already see improvement in discovery potential in the relatively simple process considered here, more complicated processes may see even greater benefit from the extra information present in the multivariate approach.

# Acknowledgements

The authors would like to thank Felix Kling, Johann Brehmer, and Kyle Cranmer for many valuable discussions throughout all stages of this project. The authors would also like to thank Benjamin Nachman and Jesse Thaler for feedback on a preliminary draft of this manuscript. This material is based upon the work supported by the National Science Foundation Graduate Research Fellowship under Grant No. DGE-1321846. DW is supported by the Department of Energy Office of Science.

# 6 Appendix

Consider a test set $\mathcal{X}_{\text{test}}$ of $N$ events drawn from the distribution $p_{test}(x)$ where $N$ is large. Using only invariant mass (denoted $m_{jj}$) to find the log likelihood ratio, we have

$$\log r(\mathcal{X}_{\text{test}}, \theta, \theta_0) = N \int dm_{jj}\, p_{test}(m_{jj}) \log r(m_{jj}|\theta, \theta_0). \qquad (6.1)$$

A binned form of the expression $\log r(m_{jj}|\theta, \theta_0)$ is plotted as the grey line in Figure 2.

Using higher dimensional observations $x = (m_{jj}, x')$ to find the log likelihood ratio, we instead have:

$$\log r(\mathcal{X}_{\text{test}}, \theta, \theta_0) = N \int dx\, p_{test}(x) \log r(x|\theta, \theta_0),$$

which can be manipulated to take the same form as Equation 6.1:

$$\log r(\mathcal{X}_{\text{test}}, \theta, \theta_0) = N \int dm_{jj}\, p_{test}(m_{jj}) \left[ \int dx'\, p(x'|m_{jj}) \log r(x', m_{jj}|\theta, \theta_0) \right]. \qquad (6.2)$$

The bracketed expression is approximated by taking the expectation within bins of invariant mass using the same binning as the previous case. This is plotted as the blue and magenta lines in Figure 2. We see that the bracketed expression is analogous to $\log r(m_{jj}|\theta, \theta_0)$ when using higher dimensional data to calculate the log likelihood ratio.

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
