# Peer review of "Resonance Searches with Machine Learned Likelihood Ratios"

_SciPost Physics_

## Round 1 · Referee Report · Anonymous (Referee 1) · 2020-4-4

Strengths

1- Applies a machine-learning-based generalization of the matrix element method to a resonance search for the first time.

2- The paper clearly describes what was done.

Weaknesses

1- There is no discussion of uncertainties. This is not just incorporating the usual nuisance parameters; for simulation-based inference, it is essential that the data actually live in the simulation manifold. A more specific suggestion is in the report.

2 - This analysis in practice uses data-driven background estimates. I did not see how this simulation-based approached meshes with a data-driven background estimate.

Report

This paper is a nice application of simulation-based inference to a resonance search in collider-based particle physics. The presentation is clear and the results adds a useful contribution to the literature. Below are some comments and suggestions that would be good to implement before this paper is published.

Major

  • You say "The full event information is given by the set of four-momenta of every particle in the final state. Summarizing this relatively high dimensional information with invariant mass plausibly results in a significant amount of information loss.", but then you say "For each event, our observables x consist of the four-vector of each reconstructed jet". So you are still only using 8 dimensions of the high-dimensional space. Can you comment on the scalability of your algorithm?

  • "We also calculate the likelihood ratio from histograms" -> do you apply any other event selection? It seems like dijet searches ofen apply some asymmetry requirements on pT and/or rapidity to select dijet s-channel events.

  • "A possible critique of this interpretation is that the histogram based approach does not perform as well as ALICE because it requires use of a binned PDF..." -> this is a nice test!

  • Thinking a little about uncertainties, it would be helpful if you could add something about the folloing:

-> What if p(x|\theta) is not correct (=nature) for any \theta?
-> What if p_\theta given in Eq. (3.1) is not correct? (if r is wrong, then your procedure may be suboptimal for precision, but this would also make your p-values wrong for accuracy)

  • Related: the dijet analysis is always done using a data-driven background estimation technique. How does that interplay with your analysis?

Minor

  • "Typical searches for BSM resonances involve fitting signal and background spectra in the invariant mass of the final state particles..." -> I know what you mean, but this is not technically correct. For example, most SUSY searches do not do bump hunts, but they are looking for BSM resonances.

  • "However, relying soley on the invariant mass..." -> 14]-[17] use more than just the invariant mass?

  • "...we can easily produce a set of events..." -> often it is rather computationally expensive to sample so maybe drop "easily"? Also, I would add the word "simulated" before "events".

  • Eq. (2.1): On the one hand, the law of total propability lets you write p(x) = \int dz p(x|z) p(z) for any z, but on the other hand, "partons" are unphysical and factorization theorems for the LHC are not exact (except for Drell-Yan) so p(x|z) and p(z) are not physical quantities. Is this relevant for your ratio in Eq. (2.2) - is there an implicit assumption that the unphysical schemes are the same for the top and bottom? You say that "z" is not well-defined in an experiment - I would say that "z" is not well-defined period (it is not observable in an experiment).

  • Your notation is a bit confusing. X is usually a random variable, but here, X is a set and x is a random variable. However, x is also used interchangeably as a realization of the random variable x. I would suggest that X be the random variable, x is the realization, and a new symbol be used for a set of events.

  • Eq. (2.9): I was a little confused when I saw this, because I thought you were saying that this is p(\theta)/p(\theta_0) because the lefthandside doesn't depend on X. Perhaps writing p(X,\theta,\theta_0) would make this clearer?

  • [37] belongs to the sentence before. It obviously doesn't matter, but why did you pick R = 0.5? No one has used that radius parameter for years now.

Requested changes

Please see the above bulleted list. Additionally, the labels in Fig. 3-5 are impossible to see - please make them relatively bigger.

---

## Round 1 · Referee Report · Anonymous (Referee 2) · 2020-4-25

Strengths

  1. The use of highdimensional event information for the construction of likelihood ratios clearly outperforms the histgram based approach.

  2. Further developpment of neural network based likelihood ratio estimates for resonances, after it was established for effective field theories.

Weaknesses

  1. The paper lacks a discussion of the limitations and liability of their approach.

Report

The present paper « Resonance Searches with Machine Learned Likelihood Ratios «  demonstrates how to increase the performance of resonance seaches by using neural networks for the estimation of likelihood ratios. The use of neural networks to learn the likelihood ratio has been demonstrated previously for EFTs. Since the matrix element can in those cases be expressed in terms of a polynomial in the Wilson coefficients, the Z’ benchmark considered in this paper represents a more complex model due to the appearance of a resonance.

The paper is well written and clearly structured. With the growing importance of machine learning methods in high energy physics, the presented performance evaluation is relevant for publication. In this context I suggest a more detailed analysis of the obtained results especially with respect to limitations of the neural networks. In particular since uncertainty estimations for neural networks are not yet established, such a discussion is necessary to estimate under which conditions the results are reliable.

Requested changes

  1. Fig. 2 illustrates the differences between the different approaches. The paper would profit from a discussion of the observed effects.

I would naively expect the dashed and the solid blue line to be the same when the distribution of events over the non displayed dimensions can not be distinguished for the chosen values of theta. However in that case I would expect those lines to coincide as well with the grey line. Starting at 350 GeV the ML method consistenly favors the $\theta_0$ hypothesis. The discrepancy in this region is larger than the modest improvement of the ML method with respect to the histogram method seen in the bulk region. I assume this is due to lower statistics in the tail, where the network has not sufficiently learned the underlying structure. Please explain the observed curves.

A horizontal line at zero would be helpful to guide the eye.

  1. Fig. 5 seems to show a much smoother behaviour compared to Fig. 4 thanks to the interpolation properties of the neural network. Is this correct?

  2. A main point of the paper is to include the full information at detector level instead of 1 dimensional information in the histogram. In the following the authors demonstrated that the 2 dimensional plane of $m_{jj}$ and $\Delta y_{jj}$ fully covers already all information contained in the two 4-dimensional jets. The authors should consider to include an extension where they could choose a more complex model to profit from the highdimensional information.

  3. The authors state that "by evaluating the squared matrix element of an event at a handful of “benchmark” points in the parameter space, one is able to infer the squared matrix element for arbitrary theory parameters." While this is true in theory, in practice the inference of the polynomial parameters - in particular of all interference terms - after showering and detector simulation requires much more than a handful of benchmark points.

  4. page 4: The sentence starting with ‘In this work’ does not seem to be complete.

---

## Round 2 · Author Response

Reviewer 1 raised some concerns over the unconverged behavior in the tails of the invariant mass distribution in Figure 2, specifically remarking that the discrepancies in this region are larger than the improvement in the bulk region. We believe it is worth mentioning that, by normalizing each bin to the number of data points within each bin, the figure exaggerates the relevance of this effect in determining the likelihood ratio of a set of events. Because significantly more events will lie in the bulk region than the tails, the improvements in this region will contribute significantly more to the likelihood ratio than the noise in the tails when summing over a set of events.
Reviewer 1 asked about the smooth behavior seen in Figure 5, compared to the relatively noisy behavior in Figure 4. As the reviewer suggested, this smoother behavior is because the neural network interpolates in theta-space, whereas the histogram approach is uninformed by the behavior at neighboring grid points.
Reviewer 1 also suggests applying this to a more complex model where we would be able to leverage the full high dimensional information. We have considered alternative, more complex models for the subject of future work. It is worth noting that our claim that all information is contained in a two dimensional subset of features can only be made as a result of the higher dimensional analysis using the full jet 4-momenta. In this way, the analysis performed in the manuscript fully leverages the available high dimensional information.
Reviewer 1 raised concerns over the sentence "By evaluating the squared matrix element...". We agree with the reviewer here that the "inverse" problem of inferring theory parameters or Wilson coefficients requires a large number of data points in spite of the morphing structure due to showering and detector simulation. Our statement here comments on the relative ease of the "forward" problem of calculating squared matrix elements (or equivalently, joint likelihood ratios) which takes place at the parton level. For a given event z, the morphing structure allows one to only evaluate the squared matrix element at some number of benchmark points in theta-space in order to infer the value at any point in theta-space. Since resonance searches do not benefit from a morphing structure due to the nonlinear dependence on mass, we must evaluate the squared matrix element of each event on a grid in theta-space in order to reproduce a similar degree of coverage.
Reviewer 2 inquired about additional event selection criterion that may have been applied to the data set. We did not apply any further cuts or selection criteria to the data set beyond those mentioned in the manuscript.
Reviewer 2 asked how we may incorporate data-driven background estimates into the analysis. One way this may be achieved is to use data-driven normalization of Monte Carlo samples to construct a partially data-driven background estimation. Joint likelihood ratios for background events, which would not possess a theta dependence, are given by a constant related to the overall cross-sections for theta and theta_0.
Reviewer 2 brings up insightful comments about Equations (2.1) and (2.2). We do not believe that the quantities p(x|z) and p(z) must necessarily be well-defined in experiment for our approach to be valuable. In Eq. (2.2), we do assume that the probability distributions p(x|z) are the same for the numerator and the denominator. However, this distribution is defined by the programs that simulate event showering and detection. These simulations will always converge to the same distribution for a given event, and so these distributions are well defined in simulation. There is of course an additional assumption that the simulation suite accurately models event generation and detection that occurs in experiment. This is a fundamental assumption that cannot be forgone, though is common in the literature and can be aided by profiling over nuisance parameters.
Reviewer 2 also brought up remarks on the notation used to denote sets of events, random variables, and events that are an instantiation of the random variable. We have changed the notation for sets of events to \Chi so that our notation is not at odds with the convention in statistics that X is a random variable and x it's realization. The notation used in our manuscript was borrowed from previous publications on these methods (specifically [23-26]). In order to not deviate from this precedence, we have added a statement clarifying that x will be used interchangeably to mean a random variable and its realization, and its meaning should be clear from context.
Reviewer 2 asked about our motivations for choosing R=.5. This choice was made so that we could check for consistency of our data with older datasets.
We would like to once again thank the reviewers for their time and consideration, and hope that we have adequately addressed their concerns and questions.

---

## Round 2 · List of Changes

We have added a horizontal line at log r = 0 in Figure 2 in order to guide the eye.
We have also added a sentence in the caption of this figure that highlights the unconverged behavior that is seen in the tail of the distribution. These changes were made in response to reviewer 1's feedback on this figure.
We have changed the sentence on page 4, which previously read "In this work, we focus initially on a qualitative assessment of the application of machine learned likelihood ratios to resonance searches." to the new sentence "We focus on a qualitative assessment of the application of machine learned likelihood ratios to resonance searches."
We have added a new sentence to the conclusion (beginning "Since these methods scale well...") which addresses the scalability of the algorithm. We have also added a citation to a high dimensional application in a different field, which uses high-dimensional images as the input data. These changes were made in response to reviewer 2's request that we comment on the scalability of the algorithm.
We have changed "Typical" to "Many" in the sentence that read "Typical searches for BSM resonances..." in the second paragraph of the introduction. This change was made in response to reviewer 2's critique of this sentence.
We have moved the citation block [14-17] in response to reviewer 2's comment from the sentence "However, relying soley on the invariant mass..." to the next paragraph, after the sentence "Certain methods have been developed to overcome this information loss."
In the sentence on page 3 which formerly read "Using a standard suite of programs, we can easily produce a set of events..." we have removed the word "easily" and included the word "simulated", which were changes suggested by reviewer 2.
We have introduced a new symbol, \Chi, to represent a set of events, which was previously represented with X.
We have changed the labels and color bars in Figures 3-5 to make them more readable.

---

## Round 3 · Author Response

We would once again like to thank the reviewer for their careful consideration of our work. The reviewer raised 2 primary concerns in their comment regarding uncertainties and background estimation. We agree with the reviewer on both of their points and have updated the manuscript in response to these concerns. Specifically, we have more clearly stated the limitations of the approach that were highlighted by the reviewer in the paragraph beginning "Built into these approaches..." on page 4.
With regards to the question of assigning uncertainties to calculated p-values, it is our understanding that this is a current topic of research surrounding these methods. Ideas for achieving this include training an ensemble of machine learning models to predict the likelihood ratio, rather than a single neural network. The ensemble mean (or alternative method of ensemble prediction) would then be the predicted likelihood ratio, and the ensemble variance could be used to quantify uncertainty in the likelihood ratio, and thus the p-values, due to the probabilistic training of ensemble members.
With regards to the question of assigning uncertainties to calculated p-values, it is our understanding that this is a current topic of research surrounding these methods. Ideas for achieving this include training an ensemble of machine learning models to predict the likelihood ratio, rather than a single neural network. The ensemble mean (or alternative method of ensemble prediction) would then be the predicted likelihood ratio, and the ensemble variance could be used to quantify uncertainty in the likelihood ratio, and thus the p-values, due to the probabilistic training of ensemble members.

---

## Round 3 · List of Changes

We have included a new paragraph beginning "Built into these approaches..." on page 4.

---

## Editorial Decision

awaiting_resubmission